# The Glucosylceramide Synthase Inhibitor PDMP Causes Lysosomal Lipid Accumulation and mTOR Inactivation

**DOI:** 10.3390/ijms22137065

**Published:** 2021-06-30

**Authors:** Pia Hartwig, Doris Höglinger

**Affiliations:** Heidelberg University Biochemistry Center (BZH), 69120 Heidelberg, Germany; pia.hartwig@bzh.uni-heidelberg.de

**Keywords:** glycolipids, sphingolipids, clickable lipids, lysosomal biology, LBPA, cholesterol

## Abstract

For many years, the biology of glycosphingolipids was elucidated with the help of glucosylceramide synthase (GCS) inhibitors such as 1-phenyl-2-decanoylamino-3-morpholino-1-propanol (PDMP). Additionally, PDMP gained interest because of its chemosensitizing effects. Several studies have successfully combined PDMP and anti-cancer drugs in the context of cancer therapy. However, the mechanism of action of PDMP is not fully understood and seems to go beyond glycolipid inhibition. Here, we used a functionalized sphingosine analogue (pacSph) to investigate the acute effects of PDMP on cellular sphingolipid distribution and found that PDMP, but not other GCS inhibitors, such as ND-DNJ (also called Miglustat), induced sphingolipid accumulation in lysosomes. This effect could be connected to defective export from lysosome, as monitored by the prolonged lysosomal staining of sphingolipids as well as by a delay in the metabolic conversion of the pacSph precursor. Additionally, other lipids such as lysobisphosphatidic acid (LBPA) and cholesterol were enriched in lysosomes upon PDMP treatment in a time-dependent manner. We could further correlate early LBPA enrichment with dissociation of the mechanistic target of rapamycin (mTOR) from lysosomes followed by nuclear translocation of its downtream target, transcription factor EB (TFEB). Altogether, we report here a timeline of lysosomal lipid accumulation events and mTOR inactivation arising from PDMP treatment.

## 1. Introduction

1-phenyl-2-decanoylamino-3-morpholino-1-propanol (PDMP) is a well-known glucosyl-ceramide synthase (GCS) inhibitor that has been used for several decades to study the effect of glycosphingolipids in a variety of biological settings [1,2,3,4]. More recently, pro-apoptotic effects of PDMP are being studied in the context of cancer therapy. Several reports show that co-treatment of PDMP increases the efficiency of anti-cancer drugs [5,6,7]. This effect was attributed to the accumulation of the glucosylceramide precursor ceramide, itself a known apoptotic agent [8,9]. However, PDMP seems to also influence other cellular lipid pathways. It has been reported to increase lysobisphosphatidic acid (LBPA), cholesterol, and lipid droplet levels [10,11,12,13]. Furthermore, it disrupts mitochondrial raft-like domains and thereby reduces mitochondrial fission events [14]. Interestingly, one report has connected cellular PDMP treatment to the inactivation of the mechanistic target of rapamycin (mTOR) in osteoblasts [15]. 

mTOR, a 289 kDa sized serine-threonine kinase, is the master regulator of multiple cellular processes such as cell metabolism, growth, proliferation, and survival [16,17,18,19]. Many stimuli such as high amino acid levels [20,21], glucose [22], and insulin [23] are known to activate mTOR, thereby promoting the biosynthesis of proteins, lipids, and organelles [16]. However, much less is known about the mechanisms of its repression. Under physiological conditions such as starvation [20], mTOR is known to be inactivated, which manifests in its dissociation from the lysosomal surface. Under these conditions, downstream targets of mTOR such as transcription factor EB (TFEB) are no longer phosphorylated and subsequently translocate to the nucleus [24]. In the nucleus, TFEB then regulates the transcription of the genes involved in autophagy and lysosomal biogenesis [25]. The recent identification of the lysosomal lipid PI(3,4)P2 as an endogenous repressor of mTOR activity marks an important first step in understanding the components involved in mTOR inactivation [26]. 

Given its importance in a great variety of cellular processes, it is not surprising that mTOR dysregulation is implicated in diseases such as cancer [27,28,29], and that synthetic inhibition of mTOR is being exploited as a therapeutic option. At this stage, two inhibitors are approved for clinical use in kidney and breast cancer [30]. 

In this study, we were curious to investigate a possible connection between PDMP treatment and mTOR inactivation, as both have been shown to promote apoptosis in tumors. The cellular actions of PDMP, beyond the inhibition of GCS, are not fully understood. This is also due to a lack of tools to follow sphingolipid metabolism in time and space. Here, we used a photocrosslinkable and clickable analogue of sphingosine (pacSph) to investigate ceramide accumulation upon PDMP treatment. Besides sphingolipids, we also analyzed other lipids, such as LBPA and cholesterol, and investigated the chronology of lipid accumulation events at the lysosome upon PDMP treatment. We further connected these events to mTOR inactivation and nuclear translocation of TFEB. Remarkably, LBPA was the first lipid to accumulate upon PDMP treatment, which concurs with mTOR dissociation from the lysosome and TFEB translocation to the nucleus. Subsequently, sphingolipids, most likely ceramide, and cholesterol accumulations could be detected in the lysosomes. 

Together, our data highlight the timeline of lipid accumulation events and mTOR inactivation arising from PDMP treatment.

## 2. Results and Discussion

### 2.1. PDMP Induces Lysosomal Ceramide Accumulation

We initially set out to characterize the effects of PDMP on sphingolipid metabolism at a subcellular level. Rather than looking at endogenous lipid levels in a steady-state situation, we were interested in how sphingolipid metabolism adapts acutely to the inhibition of GCS. To this end, we used a photoactivatable and clickable sphingosine probe (pacSph) that, when given to cells, enters endogenous lipid metabolic pathways and can later be visualized by virtue of its clickable alkyne [31]. Additionally, the metabolic fate of pacSph can be tracked through thin-layer chromatography (TLC; Figure 1A).

We compared the effects of treatment with PDMP and a structurally different GCS inhibitor, N-butyl-deoxynojirimycin (NB-DNJ; also known as Miglustat) on incorporation of the pacSph precursor into cellular lipids. We noticed that PDMP, but not NB-DNJ treatment, gave rise to higher ceramide levels compared with the untreated control (Figure 1B, quantified in 1C; for quantification of other lipids, see Appendix A). This accumulation is also seen in a sphingosine-1-phosphate lyase knock-out (SGPL1^−/−^) cell line, in which the sphingolipid exit pathway is blocked and the pacSph precursor stays within the sphingolipid cycle [32]. Interestingly, NB-DNJ treatment in SGPL1^−/−^ cells also resulted in increased conversion of ceramide, suggesting that the SGPL1-pathway is a compensatory pathway for eliminating excess sphingolipids created by blocking glycolipid conversion. In support of this, we see phosphatidylethanolamine (PE), formed downstream of SGPL1, at higher levels in control and NB-DNJ treated WT cells (Appendix A). Together, these data show that PDMP treatment increases ceramide levels through a mechanism other than its inhibitory effect on GCS, as NB-DNJ does not show such an increase. One pathway influenced by PDMP could be SGPL1-mediated cleavage of the sphingoid backbone, as NB-DNJ treatment in a SGPL1^−/−^ backgound also shows elevated ceramide levels compared with the untreated control.

Given that SGPL1 acts on phosphorylated sphingosine after its exit from the lysosomes, we next moved to investigate the subcellular distribution of pacSph-derived metabolites by confocal microscopy. To this end, we fixed pacSph-labelled cells upon UV-crosslinking and attached a fluorophore to the alkyne group by means of click chemistry. Here, PDMP-treated cells, but not control or NB-DNJ treated cells, showed large vesicular lipid accumulations (Figure 1D). Co-localization using Lysosome-associated membrane glycoprotein 1 (Lamp1) identified the vesicles as lysosomes (see Appendix A) and the quantification of co-localization as measured by Pearson’s correlation coefficient showed a significant increase of pacSph-Lamp1 co-localization in PDMP treated cells compared with the control and NB-DNJ treated cells (Figure 1E). 

Given that this lysosomal accumulation could be observed in both WT and SGPL1^−/−^ cells, we concluded the labelled lipids to be sphingolipids, most likely ceramide, as this is the only lipid to be enriched in both cell lines (Figure 1C). This ceramide enrichment upon PDMP treatment is in line with early observations that *erythro* diastereomers of PDMP, which do not inhibit GCS, also raise cellular ceramide levels [33], as well as more recent studies, which found ceramides to be enriched in dendritic cells upon PDMP treatment [9]. Our methodology allowed us to gain insight into the subcellular localization of this enrichment. The molecular mechanism for such PDMP induced ceramide accumulation is still debated. Early work already ruled out a direct action of PDMP on sphingolipid metabolizing enzymes such as ceramidase, sphingomyelinase, sphingomyelin synthase, or ceramide synthase, as the activities of these enzymes were neither stimulated nor inhibited in the presence of PDMP [34]. Even though we have not identified the exact mechanism of how PDMP influences lysosomal sphingolipid levels, we speculate that PDMP treatment could in part decrease flux through the SGPL1-pathway. However, the fact that PDMP but not NB-DNJ treatment showed a prominent lysosomal sphingolipid staining in SGPL1^−/−^ cells points to a potential additional lysosomal target of PDMP.

### 2.2. PDMP Causes Lysosomal Sphingolipid Export Defects

In order to investigate whether the observed ceramide accumulation stems from defective transport, we next used pacSph in short pulse-chase experiments. Given that short labelling with pacSph results in a predominantly lysosomal localization of the probe [31], we were able to visualize lysosomal sphingolipid export using early timepoints such as 0, 5, and 30 min.

TLC analysis of the control and PDMP treated SGPL1^−/−^ cells showed increasing metabolism of pacSph into higher sphingolipids such as ceramide and sphingomyelin over time (Figure 2A). Interestingly, PDMP treatment showed a tendency of exhibiting lower ceramide levels accompanied with slightly elevated sphingosine levels at early timepoints of 0 and 5 min (Figure 2B,C). This could indicate a possible sphingolipid export defect, as ceramide conversion is only possible once sphingosine has reached the ER following lysosomal exit. 

This seemingly contradictory result—accumulation of ceramide at long time points but a decrease at shorter ones—can be explained by the constant turnover of sphingolipids. Sphingosine outside of the lysosome is converted into higher sphingolipids (including ceramide), which are then “recycled” back to sphingosine in the lysosome. If the export of lysosomal sphingosine is impaired, its conversion to higher order sphingolipids is reduced in the short term. However, lysosomal ceramide is also the precursor of sphingosine in the catabolism of higher order sphingolipids; if those have incorporated the labelling—at long timepoints—lysosomal sphingosine and ceramide will accumulate due to impaired export.

To support that the pacSph-derived lipids are retained in the lysosomes in PDMP-treated cells, we visualized the subcellular distribution of pacSph and its metabolites using the same short pulse-chase conditions as above (Figure 2D). As reported before, the 0 min chase timepoint showed predominant lysosomal localization of the probe in both conditions (for co-localization, see Appendix A). In the control cells, this lysosomal localization was replaced by whole-cell staining within 30 min of chase, while PDMP treated cells still showed obvious lysosomal staining. Quantification of Lamp1 co-localization confirmed a significantly higher co-localization in PDMP treated cells at 30 min (Figure 2E), in line with reduced lysosomal sphingolipid export.

This could indicate a functional defect of the entire organelle. This suggestion is also supported by previous studies showing an increase in lysosomal cholesterol [11], an accumulation of multilamellar bodies, and an enrichment in the number of autophagosomes upon PDMP treatment [12]. In fact, early studies using fluorescent analogues of PDMP observed an exclusive lysosomal localization of fluorescent PDMP together with lysosomal enlargement upon overnight treatment [35].

### 2.3. PDMP Induces Lysosomal Accumulation of Multiple Lipids

To investigate how the lysosomal actions of PDMP affect other lysosomal lipid species, we next investigated its effect on lysobisphosphatidic acid (LBPA) as well as cholesterol. LBPA, also called bis(monoacylglycero)phophate (BMP), is a phospholipid exclusively found in the internal membranes of lysosome. Using fluorescent anti-LBPA antibodies as well as filipin staining, we observed a severe accumulation of LBPA and cholesterol in lysosomes in PDMP treated cells, but not in untreated control or NB-DNJ treated cells (Appendix A).

Given that multiple lipids accumulate in response to PDMP treatment, we considered whether these accumulations occur in a time-dependent manner. To this end, we incubated cells with PDMP for different timepoints (0–22 h), and analyzed lysosomal LBPA, pacSph, and cholesterol accumulation by confocal microscopy. 

LBPA was the first lipid to accumulate after 2 h of PDMP treatment (Figure 3A, quantified in B). In contrast, pacSph-derived lipid accumulation in lysosomes required 4 h of PDMP treatment (Figure 3C). Quantification of Lamp1 co-localization by Pearson’s correlation coefficient confirmed a significantly higher co-localization after 4 h (Figure 3D, for pacSph-Lamp1 co-localization see Appendix A). Interestingly, significant cholesterol enrichment occurred only after 6 h of PDMP incubation, as quantified by the filipin mean intensity (Figure 3E,F). Therefore, we hypothesize that PDMP treatment initially affects lysosomal LBPA levels whereas sphingolipid and cholesterol accumulations occur later and could be downstream of the initial LBPA accumulation. Given that lysosomal lipid levels are strictly monitored by components of the mTOR complex, we next investigated the effects of PDMP treatment and subsequent lipid accumulations on mTOR and its downstream effector TFEB.

### 2.4. PDMP Induced LBPA Accumulation Triggers mTOR and TFEB Translocation

Here, we again applied the same PDMP time-course as before in order to visualize the subcellular localization of mTOR and TFEB using immunofluorescence microscopy (Figure 4A,C).

As previously described, mTOR initially localizes to lysosomes [36] and its downstream transcrition factor TFEB is kept cytosolic [24]. However, after 2 h of PDMP treatment, the lysosomal staining disappeared and mTOR gave a predominantly cytosolic signal. Quantification of Lamp1 co-localization by Pearson’s correlation coefficient confirmed a significantly decreased co-localization after 2 h (Figure 4B; for mTOR-Lamp1 co-localization, see Appendix A). As a consequence of inactive mTOR, TFEB was no longer phosphorylated and could now enter the nucleus [37]. Consistent with mTOR inactivation, we could see nuclear translocation of TFEB after 2 h of PDMP treatment (Figure 4C). We quantified the proportion of cells showing TFEB in the nucleus. While at 0 h, all of the cells contained only cytosolic TFEB, and at 2 h, we could already observe 10% of cells with a nuclear signal, and that proportion increased to 35% within 6 h of PDMP treatment (Figure 4D; for co-localization with DAPI, see Appendix A).

Both mTOR and TFEB changed their subcellular localizations following 2 h of PDMP treatment. As LBPA was the only investigated lipid to accumulate in that timeframe, we hypothesize that PDMP-induced LBPA accumulation could be a trigger for mTOR inactivation and dissociation from the lysosomes. However, the exact mechanism of this inactivation is still unclear and the question whether this involves a direct action of LBPA on components of the mTOR complex will be subject of further study.

## 3. Conclusions

In conclusion, the presented study emphasizes a timeline of lipid accumulation events and mTOR inactivation arising from PDMP treatment. While PDMP is classically thought of simply as a glycolipid synthesis inhibitor, its phenotypic consequences are very different to those of newer generation inhibitors such as NB-DNJ. Indeed, by following the metabolic fate of a functionalized sphingosine analogue in a time-resolved manner, we were able to see that PDMP leads to a different set of lipid levels than NB-DNJ, particularly the specific accumulation of ceramide. As this phenotype could be partially reproduced with NB-DNJ in a SGPL1^−/−^ cell-line, we speculate that ceramide accumulation resulting from GCS inhibition might compensated for via the SGPL1-pathway in a wild type cell. However, the mechanistic details of this cascade, particularly which enzymes are directly affected, remain unclear, and will need to be investigated further. Moreover, PDMP has an additional effect on lysosomal functions, as shown by defective sphingolipid export and the accumulation of both LBPA and cholesterol.

We were able to pinpoint early LBPA enrichment and subsquent mTOR inactivation as potential upstream events of these lipid accumulations. Although not mechanistically explored in this manuscript, it is exciting to speculate the LBPA could act, alongside the recently discovered actions of PI(3,4)P2 [26], as an endogenous repressor of mTOR activity. Altogether, our findings on the early actions of PDMP on lysosomes will be important to consider not only in the context of inducing tumor cell apoptosis, but more generally when interpreting data from glycolipid-inhibition experiments using PDMP.

## 4. Materials and Methods

### 4.1. Cell Culture

HeLa WT and sphingosine-1-phosphate lyase kock-out (SGPL1^−/−^) cells were cultured in DMEM supplemented with 100 U/mL Penicillin-Streptomycin (Thermo Fischer Scientific) and 10% FBS (Bio and Sell, Feucht, Germany). HeLa SGPL1^−/−^ cells in which SGPL1 was knocked out using CRISPR/Cas9 genome editing were kindly provided by Prof. Dr. Britta Brügger (Biochemistry Center, Heidelberg University). The cloning strategy, the respective oligonucleotides, and the transfection procedure can be found in M. J. Gerl et al., 2016, PLoS One [32].

### 4.2. Antibodies

Mouse anti-LBPA (MABT837) antibody was from Millipore, Merk (diluted 1:100), and the rabbit anti-Lamp1 (#9091), mouse anti-Lamp1 (#15665), rabbit anti-mTOR (#2983), and rabbit anti-TFEB (#4240) were purchased from Cell Signalling (diluted 1:200, 1:100, 1:100, and 1:100, respectively). 

### 4.3. Inhibitors

Cells were treated for indicated times with 20 µM PDMP (Sigma Aldrich, Darmstadt, Germany; 10 mM stock in ethanol) or 50 µM NB-DNJ (kind gift from Prof. Fran Platt, University of Oxford; 20 mM stock in DMSO) in DMEM. In the pacSph experiments, the labelling solution containing pacSph also contained the respective inhibitor at the same time.

### 4.4. Visualization of Clickable and Photocrosslinkable Sphingosine (pacSph) in Cells

Cells were seeded onto 11 mm coverslips placed in wells of a 24-well plate to 65–75% confluency. Inhibitors were added as described, and the cells were labelled with 2 µM pacSph in serum-free DMEM for the indicated times. Subsequently, cells were washed with 1 mL PBS three times at RT. Cells were overlaid with 0.5 mL of cold imaging buffer (20 mM HEPES, 115 mM NaCL, 1.2 mM MgCl_2_, 1.2 mM glucose and 1.8 mM CaCl_2_, pH 7.4/NaOH), and UV-irradiated (λ~365 nm) on ice for 5 min. Cells were immediately fixed with pre-cooled MeOH at −20 °C for 20 min. Non-cross-linked lipids were extracted by washing three times with 0.78 mL of CHCl_3_/MeOH/AcOH (10:55:0.75) (*v*/*v*) and twice with PBS. Cells were then incubated with 50 μL of click mixture (1 µL 2 mM Alexa-555-azide, 125 µL 10 mM Cu(I)BF_4_ in acetonitril and 0.5 mL PBS) for 1 h at room temperature in the dark. Cells were then washed with PBS and incubated with 50 μL of primary α-LAMP1 antibody (1:200 in PBS supplemented with 2% BSA and 0.3% Triton X-100) for 1 h in the dark. Coverslips were briefly washed with PBS and incubated with secondary antibody (α-rabbit conjugated to Alexa Fluor 488, 1:800) for 30 min, washed briefly with PBS, and mounted in ProLong Gold Antifade mounting medium (Cell Signalling, Danvers, MA, USA). Microscopy images were captured using a confocal laser-scanning microscope (Zeiss LSM800) with a 63 × oil objective.

### 4.5. Thin-Layer Chromatographic Analysis of Clickable and Photocrosslinkable Sphingosine (pacSph)

Cells were grown in 12-well plates to 85–95% confluency. Inhibitors were added as described, and the cells were labelled with 2 µM pacSph in serum-free DMEM for indicated times. The cells were washed three times with PBS, and were trypsinized and transferred into a 1.5 mL tube. After transfer, the cell solution was centrifuged (1500 rpm, 5 min) and the supernatant was discarded. Cells were resuspended in 300 µL PBS and mixed with 600 µL MeOH and 150 µL CHCl_3_ to precipitate the proteins and DNA. The mixture was vortexed and centrifuged (14,000 rpm, 5 min) and the lipid containing supernatant was transferred into a 2 mL vial. Then, 300 µL CHCl_3_ and 600 µL acetic acid (0.1% *v*/*v*) were added, the mixture was vortexed and centrifuged (14,000 rpm, 5 min) again and the aqueous phase was discarded. The organic phase was transferred into a 1.5 mL vial and dried in a speed-vac (30 °C, 20 min). The lipids were dissolved in a 30 µL click-mixture (0.6 µL, 44.5 mM 3-azido-7-hydroxycoumarin, 125 µL 10 mM Cu(I)BF_4_ in acetonitrile and 0.5 mL EtOH). Click reaction was performed in a speed-vac (45 °C, 20 min). Clicked lipids were dissolved in 15 µL EtOH/ACN (5:1) and were applied on a 10 × 20 cm TLC Silica gel 60 aluminium plate. TLC plates were developed using CHCl_3_/MeOH/H_2_O/AcOH (65:25:4:1) for 5 cm and then cyclohexane/ethylacetate (1:1) for 9 cm. Lipids containing the fluorescent coumarin group were visualized by UV light using a geldoc system, and the single lipid species were identified using commercially available pac- or clickable standard lipids.

### 4.6. Immunofluorescence Microscopy

Cells were grown onto 11 mm coverslips in a 24-well plate to be 65–75% confluent. They were fixed in 4% formaldehyde for 20 min at RT and were subsequently rinsed twice with PBS. Formaldehyde was quenched with 20 mM glycine in PBS for 10 min at RT and washed again with PBS. The first antibody was added in 1% BSA/0.3% Triton/PBS and was incubated for 1 h at RT. Coverslips were washed briefly with PBS, and a secondary antibody was added (1:800, Alexa Fluor 488 or Alexa Fluor 594 or Alexa Fluor 647, Cell Signalling) in the same antibody dilution solution and was incubated for 30 min at RT. Coverslips were washed with PBS and were mounted in ProLong Gold Antifade mounting medium (Cell Signalling). Microscopy images were captured using confocal microscopy (Zeiss LSM800) with a 63 × oil objective.

### 4.7. Image Analysis

Images were analyzed on Fiji (W. Rasband, NIH, Bethesda, MD, USA) using the FluoQ macro for automatic extraction [38] of intensity values. Co-localization was analyzed with the Coloc2 tool (https://imagej.net/Coloc_2 (accessed on 1 May 2021)) using the Costes threshold regression. These values were subsequently loaded in R and grouped according to conditions. Graphs were generated using the ggplot2 package in R [39].

### 4.8. Statistical Analyses 

All of the biochemical data were from biological triplicates, unless otherwise indicated, whereas imaging data were quantified from multiple cells, as indicated in the figure legends. Statistical analysis was performed with a Welch two sample *t*-test or a Chi-squared test for comparison of the two groups using R. Significance is indicated using asterisks (**** *p*  ≤  0.0001, *** *p*  ≤  0.001, ** *p*  ≤  0.01, * *p*  ≤  0.05, and N.S. *p* > 0.05).

## Figures and Tables

**Figure 1 ijms-22-07065-f001:**
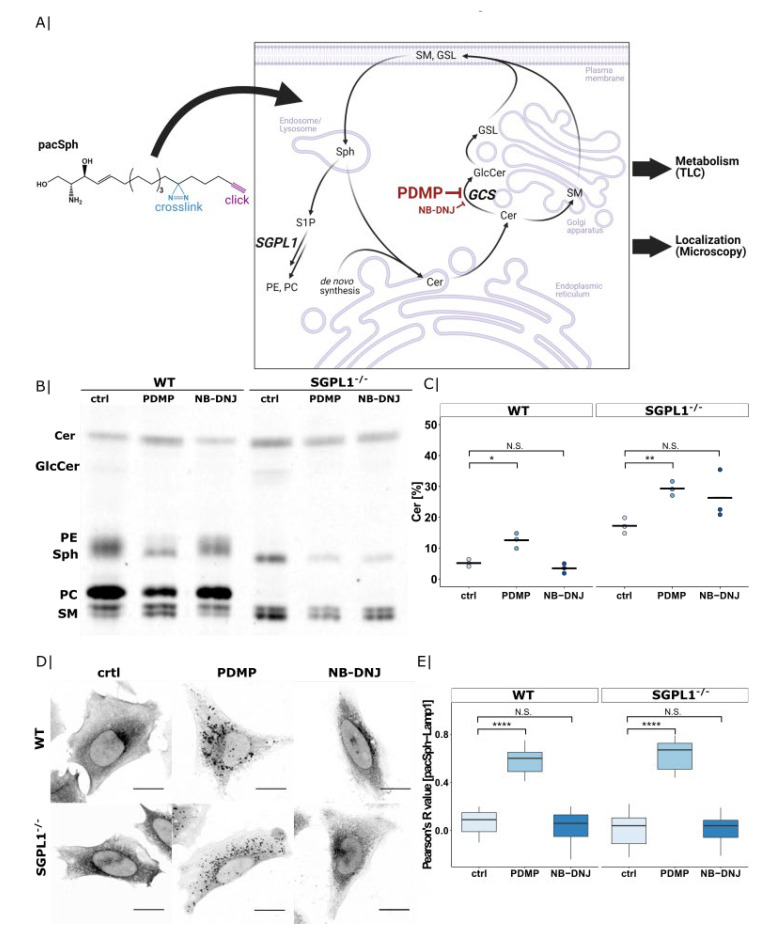
Ceramide accumulation inside lysosomes in HeLa WT and sphingosine-1-phosphate lyase knock-out (SGPL1^−/−^) cells upon 1-phenyl-2-decanoylamino-3-morpholino-1-propanol (PDMP) treatment. (**A**) Chemical structure of pacSph with highlighted functional groups and schematic illustration of its application in metabolism and subcellular localization studies. Created with BioRender.com (accessed on 1 May 2021). (**B**) Thin-layer chromatography (TLC) of pacSph metabolism in HeLa WT and SGPL1^−/−^ cells treated with PDMP (20 µM) or NB-DNJ (50 µM) for 22 h. Cells were labelled with pacSph (2 µM for 3 h) and lipids were extracted and clicked to a fluorogenic coumarin-azide. Sphingolipid species were separated by TLC. Cer—ceramide; GlcCer—glucosylceramide; PE—phosphatidylethanolamine; Sph—sphingosine; PC—phosphatidylcholine; SM—sphingomyelin. (**C**) Quantification of ceramide-levels in Figure 1B. The fluorescent signal corresponding to the ceramide-band was divided by the total intensity of all fluorescently labelled lipids. This was extracted from three independent experiments and is presented as a dot plot including the calculated mean. Welch two sample *t*-tests were performed between the control and PDMP or NB-DNJ conditions (WT ctrl-PDMP: * *p* = 2.1 × 10^−2^, WT ctrl-NB-DNJ: N.S. *p* = 2.1 × 10^−1^, SGPL1^−/−^ ctrl-PDMP: ** *p* = 3.7 × 10^−3^, SGPL1^−/−^ ctrl-NB-DNJ: N.S. *p* = 1.8 × 10^−1^). (**D**) Confocal microscopy images of pacSph-derived lipids. HeLa WT and SGPL1^−/−^ cells were treated with PDMP (20 µM) or NB-DNJ (50 µM) for 22 h, continuously pulsed with pacSph (2 µM for 3 h), cross-linked by UV-irradiation and clicked with Alexa555-azide. Scale bar indicates 20 µm. (**E**) Quantification lysosomal localization of pacSph-derived lipids. Pearson’s R value between sphingolipids and immunofluorescence signale of organelles stained with Lamp1 was extracted for each inhibitory condition (WT ctrl: *n* = 10 cells, WT PDMP: *n* = 17 cells, WT NB-DNJ: *n* = 13 cells, SGPL1^−/−^ ctrl: *n* = 11 cells, SGPL1^−/−^ PDMP: *n* = 10 cells, SGPL1^−/−^ NB-DNJ: *n* = 12 cells) and presented as boxplots. Center lines show the median, box limits indicate first (Q1) and third quartiles (Q3), whiskers extend to a maximum distance of 1.5*IQR (interquartile range) from Q1 and Q3, respectively, or to the most extreme data point within that range. Welch two sample *t*-tests were performed between the control and PDMP or NB-DNJ conditions (WT ctrl-PDMP: **** *p* = 8.1 × 10^−17^, WT ctrl-NB-DNJ: N.S. *p* = 9.6 × 10^−1^, SGPL1^−/−^ ctrl-PDMP: **** *p* = 3.5 × 10^−10^, SGPL1^−/−^ ctrl-NB-DNJ: **** *p* = 6.7 × 10^−1^).

**Figure 2 ijms-22-07065-f002:**
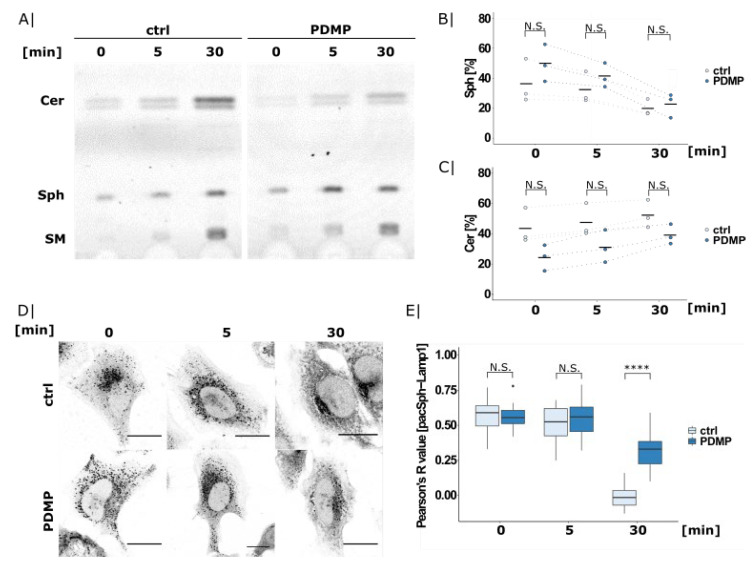
pacSph export from lysosomes studied by short pulse-chase experiments. (**A**) TLC analysis of pacSph metabolism in SGPL1^−/−^ cells. Cells were treated with PDMP overnight (20 µM) and labelled with 2 µM pacSph for 5 min. After chasing in lipid-free medium for 0, 5, or 30 min, the cells were harvested, and the lipids were extracted and clicked to a fluorogenic coumarin-azide. Sphingolipid species were separated by TLC. Quantification of (**B**) sphingosine and (**C**) ceramide levels. The fluorescent signal corresponding to the sphingosine or ceramide band from the TLC experiment shown in Figure 2A was divided by the total intensity of all of the fluorescently labelled lipids. This was extracted from three independent experiments and is presented as a dot plot, including the calculated mean. Welch two sample *t*-tests were performed between the control and PDMP (0 min: N.S. *p* = 2.9 × 10^−1^; 5 min: N.S. *p* = 3.1 × 10^−1^; and 30 min: N.S. *p* = 6.4 × 10^−1^). (**D**) Confocal microscopy images of PDMP-treated and pacSph-labelled cells as described above. Lipids were cross-linked and clicked to a fluorophore upon fixation. Lysosomes were visualized using Lamp1 antibody staining (Appendix A). Scale bar indicates 20 µm. (**E**) Pearson’s correlation coefficient of pacSph and Lamp1 was extracted for each condition (0 min ctrl: *n* = 25 cells; 5 min ctrl: *n* = 12 cells; 30 min ctrl: *n* = 20 cells; 0 min PDMP: *n* = 24 cells; 5 min PDMP: *n* = 18 cells; and 30 min PDMP: *n* = 23 cells) and illustrated as boxplots. Center lines show median; box limits indicate first (Q1) and third quartiles (Q3); and whiskers extend to a maximum distance of 1.5*IQR (interquartile range) from Q1 and Q3, respectively, or to the most extreme data point within that range. Welch two sample *t*-tests were performed between control and PDMP (0 min: N.S. *p* = 7.7 × 10^−1^; 5 min: N.S. *p* = 5.5 × 10^−1^; and 30 min: **** *p* = 1.3 × 10^−11^).

**Figure 3 ijms-22-07065-f003:**
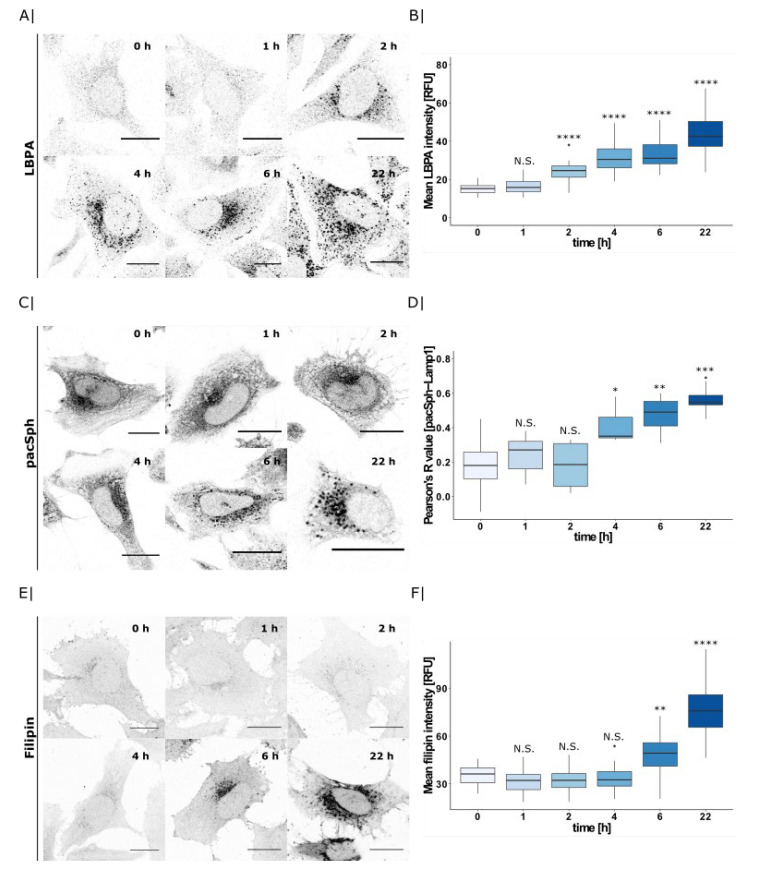
PDMP time-course and its effect on lysosomal lipid accumulation. (**A**) HeLa WT cells were treated with PDMP (20 µM) for 0–22 h and LBPA was visualized by immunofluorescence using an LBPA-antibody. Scale bar indicates 20 µm. (**B**) Mean LBPA intensity was calculated for each cell (0 h: *n* = 57 cells, 1 h: *n* = 44 cells, 2 h: *n* = 25 cells, 4 h: *n* = 36 cells, 6 h: *n* = 30 cells, and 22 h: *n* = 37 cells) and are presented as boxplots. Center lines show the median; box limits indicate first (Q1) and third quartiles (Q3); and whiskers extend to a maximum distance of 1.5*IQR (interquartile range) from Q1 and Q3, respectively, or to the most extreme datapoint within that range. Welch two sample *t*-tests were performed between 0 h timepoint and all other timepoints (1 h: N.S. *p* = 5.8 × 10^−2^, 2 h: **** *p* = 6.9 × 10^−9^, 4 h: **** *p* = 1.5 × 10^−16^, 6 h: **** *p* = 3.0 × 10^−15^, 22 h: **** *p* = 5.7 × 10^−19^). (**C**) Confocal microscopy images of sphingolipids and lysosomes. HeLa SGPL1^−/−^ cells were treated with PDMP (20 µM for 0–22 h), pulsed with pacSph (2 µM for 5 min), and afterwards were chased for 15 min in a medium without lipids. Subsequently, SL were crosslinked by UV-irradiation. Crosslinked pacSph were clicked to Alexa555-azide and lysosomes were visualized using immunofluorescence staining against Lamp1. Scale bar indicates 20 µm. (**D**) Co-localization was quantified by calculating Pearson’s R value between pacSph and Lamp1 for each timepoint (0 h: *n* = 15 cells, 1 h: *n* = 11 cells, 2 h: *n* = 14 cells, 4 h: *n* = 12 cells, 6 h: *n* = 13 cells, and 22 h: *n* = 11 cells) and are presented as boxplots. Center lines show the median; box limits indicate first (Q1) and third quartiles (Q3); and whiskers extend to a maximum distance of 1.5*IQR (interquartile range) from Q1 and Q3, respectively, or to the most extreme datapoint within that range. Welch two sample *t*-tests were performed between the 0 h timepoint and all of the other timepoints (1 h: N.S. *p* = 3.7 × 10^−1^, 2 h: N.S. *p* = 9.8 × 10^−1^, 4 h: * *p* = 1.2 × 10^−2^, 6 h: ** *p* = 1.2 × 10^−3^, and 22 h: *** *p* = 2.0 × 10^−4^). (**E**) Confocal microscopy images of HeLa WT cells treated with PDMP (20 µM for 0–22 h) and stained with 50 µg/mL filipin to visualize free cholesterol. Scale bar indicates 20 µm. (**F**) Mean filipin intensity was calculated for each cell (0 h: *n* = 41 cells, 1 h: *n* = 39 cells, 2 h: *n* = 42 cells, 4 h: *n* = 43 cells, 6 h: *n* = 50 cells, and 22 h: *n* = 54 cells) and are presented as boxplots. Center lines show the median; box limits indicate first (Q1) and third quartiles (Q3); and whiskers extend to a maximum distance of 1.5*IQR (interquartile range) from Q1 and Q3, respectively, or to the most extreme datapoint within that range. Welch two sample *t*-tests were performed between 0 h timepoint and all other timepoints (1 h: N.S. *p* = 9.7 × 10^−2^, 2 h: N.S. *p* = 6.6 × 10^−1^, 4 h: N.S. *p* = 6.8 × 10^−1^, 6 h: ** *p* = 2.6 × 10^−3^, and 22 h: **** *p* = 2.8 × 10^−14^).

**Figure 4 ijms-22-07065-f004:**
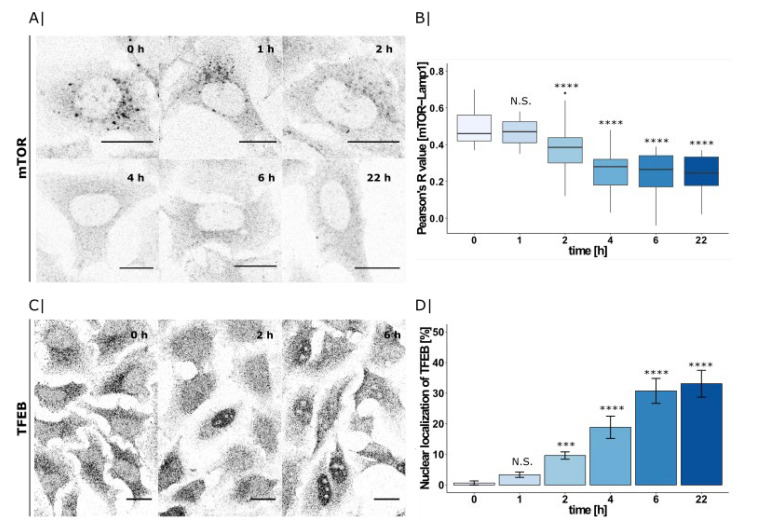
PDMP time-course and its effect on mTOR and TFEB translocation. (**A**) HeLa WT cells treated with PDMP (20 µM) for 0–22 h and mTOR visualized by immunofluorescence using mTOR-antibody. Scale bar indicates 20 µm. (**B**) Co-localization with Lamp1 was quantified by calculating Pearson’s R value between mTOR and Lamp1 for each timepoint (0 h: *n* = 29 cells, 1 h: *n* = 28 cells, 2 h: *n* = 46 cells, 4 h: *n* = 41 cells, 6 h: *n* = 32 cells, and 22 h: *n* = 32 cells) and presented as boxplots. Center lines show the median; box limits indicate first (Q1) and third quartiles (Q3); and whiskers extend to a maximum distance of 1.5*IQR (interquartile range) from Q1 and Q3, respectively, or to the most extreme datapoint within that range. Welch two sample *t*-tests were performed between 0 h and all of the other timepoints (1 h: N.S. *p* = 2.4 × 10^−1^, 2 h: **** *p* = 1.1 × 10^−5^, 4 h: **** *p* = 2.4 × 10^−12^, 6 h: **** *p* = 2.6 × 10^−12^, and 22 h: **** *p* = 5.7 × 10^−15^). (**C**) Immunofluorescence microscopy of TFEB in HeLa WT cells treated with PDMP (20 µM) for 0 and 2 h. Scale bar indicates 20 µm. (**D**) Percentage of cells with nuclear staining calculated for each time point (0 h: *n* = 155 cells, 1 h: *n* = 179 cells, 2 h: *n* = 141 cells, 4 h: *n* = 122 cells, 6 h: *n* = 143 cells, and 22 h: *n* = 142 cells) and presented as bargraphs. Error bars show the standard error of the mean. For significance testing, Chi-square tests were performed using cell counts between 0 h and all other timepoints (1 h: N.S. *p* = 1.5 × 10^−1^, 2 h: *** *p* = 4.0 × 10^−4^, 4 h: **** *p* = 2.9 × 10^−7^, 6 h: **** *p* = 1.3 × 10^−12^, and 22 h: **** *p* = 1.1 × 10^−13^).

## Data Availability

Microscopy raw data are available from ‘Biostudies’ repository with the accession number ‘S-BIAD144’.

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
