# Peer review of "The Glucosylceramide Synthase Inhibitor PDMP Causes Lysosomal Lipid Accumulation and mTOR Inactivation"

_ijms, 2021, doi:10.3390/ijms22137065_

Round 1
Reviewer 1 Report
The authors have used pac-Sph, a bifunctional sphingosine derivative and clickable photoaffinity label, to study the influence of PDMP on cellular lipid metabolism and mTOR activation.
PDMP is known as inhibitor of glucosylceramide synthase (GCS), but when the authors compared it to another GCS inhibitor, NB-DNJ / miglustat, they observed substantial differences in staining with pacSph.
Main results were that PDMP induced lysosomal ceramide accumulation, impaired export of sphingolipids from lysosomes, induced accumulation of lysobisphosphatidic acid (LBPA) and cholesterol, and finally triggered translocation of mTOR and its target transcription factor, TFEB. Translocation of TFEB to the nucleus suggested inactivation of mTOR, and the authors suggest that, because of the specific time courses of the events, LBPA could be the relevant factor for mTOR inactivation. They furthermore suggest that PDMP caused a decreased flux through the SGPL1 pathway.
PDMP is an old and widely used inhibitor of GCS, and the study may contribute to unravel its off-target effects. Overall, it is a thorougly performed study and the quality of the experiments appears to be very high. However, there are several open questions and the conclusions are not always supported by the data.
- Figure 1D: What is the evidence that pacSph-labelling in PDMP-treated cells indeed identifies ceramide? The TLC plate shows that there are other lipids that are quite strongly labelled, and SM is also significantly increased after PDMP treatment (Fig. S1C).
- Fig. S1A: PDMP has no significant effect on GlcCer, in contrast to NB-DNJ. How can this be explained?
- Fig. 2D and page 6 line 183: "0 min chase timepoint showed predominant lysosomal localization of the probe in both conditions" - however, 0 min control cells show a major staining of a compartment localized at one side of the nucleus, suggestive of Golgi.
- If PDMP really decreases the flux through the SGPL1 pathway, does it inhibit ceramidase, sphingosine kinases or SGPL1? Is anything known about it, or can the authors show this?
- Can the authors show that LBPA is in fact the lipid that directly affects mTOR?
Points 4 and 5 are of particular importance because they would strengthen very much the otherwise weakly supported conclusions of the study.
Minor point: On the TLC plates, how were standard lipids identified? Were standard lipids labelled with pac-Sph before TLC?
Reviewer 2 Report
- Please improve the introduction by adding about mTOR functions and its pathway, the ref. #16. Also Please define how the TFEB would be involve in your strategy and your evaluation.
- Please match your fig. legends and your figs labeling.
- There is no definition of SGPL gene and no protocol about the KO strategy.
- All confocal figures are not reliable since you provided black and white pictures. validating your claim about colocalization is not possible. How many cells were evaluated in each confocal experiment?
- On page 6 the authors concluded the defect of sphingolipid lysosomal exit only based on TLC data, fig 2A data, which is not a sufficient data.
- The conclusion does not support enough the authors hypothesis regarding mTOR and TFEB inactivation.
- Figure, S2, S3, S5 and S6 need quantification analysis and they do not show colocalization.
Round 2
Reviewer 1 Report
The relevant questions remain unanswered. The authors did not identify the enzymes that are relevant for PDMP-induced lipid accumulation. Their assumption that ceramides are the sphingolipids that accumulate in lysosomes is based on indirect evidence. Thus, although the methodology is highly innovative and the technical quality of the study is high, the significance of the conclusions is rather limited.
Author Response
We agree with the reviewer that a direct mechanistic link between PDMP and (sphingo)lipid accumulation could not be shown in this study. On the other hand, some lipid accumulation effects of PDMP are known for decades (first reports date back to the 1980s) and to this date, no clear mechanism of these “off-target” effects of PDMP could be revealed. We believe our study adds to the already existing body of work by providing: i) superior spatio-temporal resolution of the different lipid accumulation events, ii) the ability to track sphingosine (and their metabolic conversions to higher sphingolipids) as well as iii) the chronology of lipid accumulations in relation to inactivation of the mTOR complex.
To reflect this lack of mechanistic insight, we have further toned down our conclusions by adding (lines 148-150):
“Even though we have not identified the exact mechanism of how PDMP influences lysosomal sphingolipid levels, we speculate that PDMP treatment could in part decrease flux through the SGPL1 pathway.”
and in the conclusion section (lines 307-309):
“However, the mechanistic details of this cascade, particularly which enzymes are directly affected, remain unclear and will need to be investigated further.”
Reviewer 2 Report
It would be more informative if authors add in the abstract;
1- The Miglustat as the other compound in their study.
2- Explain in a short statement about "lysosomal export defects"
please include the name of SGPL-/- cell line in your manuscript.
Author Response
We thank the reviewer for the suggestions. We have optimized the abstract accordingly and added (lines 14-17):
“Here we use a functionalized sphingosine analogue (pacSph) to investigate the acute effects of PDMP on cellular sphingolipid distribution and found that PDMP, but not other GCS inhibitors, such as NB-DNJ (also known as Miglustat), induces sphingolipid accumulation in lysosomes.”
And rewrote the sentence previously containing “lysosomal export defects” (lines 18-20):
“This effect could be connected to defective export from lysosome as monitored by prolonged lysosomal staining of sphingolipids as well as by a delay in metabolic conversion of the pacSph precursor.”
We also included the name of SGPL1-/- cell line in the manuscript (lines 87 and 321).
Additionally, we have further toned down our conclusions by adding (lines 148-150):
“Even though we have not identified the exact mechanism of how PDMP influences lysosomal sphingolipid levels, we speculate that PDMP treatment could in part decrease flux through the SGPL1 pathway.”
and in the conclusion section (lines 307-309):
“However, the mechanistic details of this cascade, particularly which enzymes are directly affected, remain unclear and will need to be investigated further.”